# Provenancing 16<sup>th</sup> and 17<sup>th</sup> century CE building timbers in Denmark–combining dendroprovenance and Sr isotopic analysis

**Alicia Van Ham-Meert**[1,2]*, **Aoife Daly**[1,3]

**1** Saxo Institute, University of Copenhagen, Copenhagen, Denmark, **2** Department of Geosciences and Natural Resource Management (Geology Section), University of Copenhagen, Copenhagen, Denmark, **3** Dendro.dk, Copenhagen V, Denmark

☉ These authors contributed equally to this work.
* alicia.vanhammeert@kuleuven.be

**Data Availability Statement:** All relevant data are within the manuscript and its Supporting Information files.

## Abstract

Dendrochronology (tree-ring analysis) allows us to precisely date and identify the origin of timber from historic contexts. However, reference datasets to determine the origin can include timber of non-local origin. Therefore, we have applied Sr isotopic on timbers from three buildings in Jutland, Denmark, mostly dating from the late 16<sup>th</sup> and early 17<sup>th</sup> centuries CE to improve and refine the provenance identification. The dendrochronology suggested that some timbers analysed were imported from the Swedish side of Øresund/Kattegat while others were local, and others again might be from south Norway. By adding the Sr isotopic analysis, a far more detailed interpretation of the origin of these timbers can be presented for non-Danish timbers. In this paper we suggest that Danish ports in the provinces of Halland and Skåne played a major role in the timber trade between the Danish and Swedish parts of the Danish kingdom. For Danish timbers dendroprovenancing proved better than Sr isotopic analysis. Furthermore, a small number of Sr isotopic analyses were performed to contribute to the base-line along the Göta-river in Southern-Sweden.

## Introduction

### Timber trade in southern Scandinavia

Timber has been an essential building material throughout the human past. Exploitation of this resource in times of economic growth impacted the availability of bulk timber in some regions. Even in the Roman period evidence of long-distance timber transport has been demonstrated (e.g. [1–3]). But in northern Europe after the fall of the Roman Empire the transport of this bulk commodity ceased, and for almost a millennium, long distant movement of timber as a raw material is the exception. Gradually however, from around the mid-14th century CE onwards, wood and timber in Northern Europe was traded from regions with more abundant forests to regions where these materials were in high demand and no longer locally available (e.g. [4–11]). As ships became larger, carrying capacity for larger timber volumes also increased. Through dendrochronological analysis of the remains of timber structures, objects

**Funding:** AD received an starting grant from the European Research Council under the European Union's Horizon 2020 research and innovation program (Grant agreement No. 677152). Their website is: https://erc.europa.eu/funding/starting-grants. The funders had no role in study design, data collection and analysis, decision to publish, or preparation of the manuscript.

**Competing interests:** The authors have declared that no competing interests exist.

that survive in historic buildings and collections, and that are found through archaeological excavation, a detailed mapping of the sources and destinations for timber has been possible. As these structures are analysed dendrochronologically a rigorous chronological framework for past trade of timber as a raw material and trade of movable wooden objects is emerging.

Master and site chronologies for many regions, from Ireland to Estonia, from Norway to Spain, that have been built from thousands of tree-ring analyses over many decades of dendrochronology in Europe, are the tool for identifying the region of origin of historic timber (e.g. [12–20]) and summarised in [21]. However, due to the highly mobile nature of this material (import of timber, timbers as part of moving/trade objects like ships, artworks, barrels etc.) chronologies built from past structures do not necessarily reflect material from the hinterland of a site. Rather they represent a composite image of imported and local material. Over time in some regions, depending on opportunity or demand, the balance between imported and local material changed. The terrestrial dataset for dendroprovenance therefore needs to be interrogated to identify local versus imported material. The goal is not only to obtain detailed knowledge of past timber resource procurement but also to develop a more precise tool for identifying the origin of historical timber. This is a particular challenge for studies of northern European timber remains from after c. 1550 CE, when movement of timber across regions increased enormously. Timber found in towns in Denmark, for example, were often imported from other regions of Scandinavia (e.g. [22, 23]). This phenomenon is difficult to identify based solely on dendrochronological information.

Timber was increasingly traded across northern Europe. From the mid-14th century onwards, oak was shipped from the regions south and east of the Baltic Sea, in the form of boards of varying sizes. This Baltic trade continues for three centuries [24]. Oak coming from this region is almost exclusively in the form of staves, boards or planks or other converted oak, never as bulk timber. From the mid-16th century CE, bulk timber is traded in increasing volumes: Swedish timber appears in Scotland [25], Norwegian timber also reaches Scotland [10], The Netherlands (e.g. [26] and Denmark [27, 28].

In dendrochronological analysis of historical and archaeological timber remains, due to the frequent appearance of oak in Denmark that seems to be from the Swedish side of the Øresund (the sound between current-day Sweden and Denmark), it can be difficult to use this dataset to identify the geographical origin of timber. Take for example the analysis of one of the ships found during excavations in Oslo harbour (e.g. [29]). The dendrochronological analysis of the ship remains of Barcode 14 indicate that this vessel, built in c. 1574 CE, was made from oak that grew in southern Scandinavia [22]. The tree-ring series from this boat correlates with a range of sites with timbers that might come from the Swedish side of Øresund, but confirming this assertion needs to be checked independently. Though we can presume part of the material is Swedish and part Danish and have ideas of which one is which it is impossible to prove these attributions dendrochronologically in the absence of confirmed Swedish equivalents.

In the framework of the TIMBER ERC project Strontium (Sr) isotopic analysis was explored to complement this dendrochronological data. If an indication of geological origin for these trees could be attained through Sr analysis, the issue of which of the dendrochronological groups are Swedish timber could be solved. Extensive tests for Sr isotopic analysis of timbers from shipwrecks and other waterlogged wood were carried out in this project, and by others [30–33]. Unfortunately, Sr isotopic analysis of waterlogged wood has proven impossible [30–33]. However, the authors had access to timber samples from several historic buildings. These had never been submerged, hence any Sr in the wood should reflect the geology of the place where these trees originally grew. A selection of these showed, through dendrochronology, to have different provenances, some matching best with Danish datasets (early 17th century), others of probable Swedish origin (mid-16th century CE). This will allow to test the

hypothesis as to whether or not Sr isotopic analysis of non-waterlogged archaeological could be used for provenancing and complement dendrochronological results.

## Sr isotopes of (construction) timbers

In recent years narratives on wood selection and use have grown increasingly important in dendroarchaeology [34], moving beyond dating material to answer deeper questions on human choice, agency and trade (e.g. [7, 35]). This movement has allowed new questions to be asked, new venues of research to be explored and new layers of understanding to be added to the historical and archaeological picture. This leads to the inclusion of archaeological timbers that cannot be dated and to innovative techniques being used to complement the picture such as aDNA analysis, elemental and isotopic analysis (O,N,C and Sr). These lead to more holistic information on landscape use, forestry management, wood and timber exploitation etc. As the excitement subdues and limitations appear, the end of those methods is not in sight but rather a more holistic approach is to be championed, where archaeologically driven questions make use of these methods, rather than methodological developments driving the questions. A helpful review of recent developments both in the theoretical framework and methodologically is offered by Domínguez-Delmás [34].

Provenancing of ancient timbers is most often achieved, much like dating, through statistical matching of tree-ring sequences with existing chronologies. When this is not possible species identification (visually, by computer-aided image recognition, through chemometrics or aDNA) can provide an alternative means of provenancing if the (sub-)species is present only in certain locations and that this information, combined with archaeological information, provides a decisive answer [34]. Seeing the recent explosion in the use of Sr isotopes to study human mobility [36] the relative scarcity of Sr isotopic studies of timbers is somewhat surprising (see Table 1). It can, in part, be attributed to alternative means of provenancing available for timbers through dendroprovenancing. Another, possible answer is that the interpretation of Sr isotopic results is not as straightforward as first expected. Obtaining a signature comes with challenges in the form of exogenous material. As has been shown recently, Sr isotopic analysis of waterlogged wood is not possible due to the influence of Sr present in the waterlogging environment which cannot be removed [31, 32]. This seriously diminishes the potential material available for Sr isotopic study of wood. Even admitting one obtains the "real" signature, this signature on its own will not allow to pinpoint the origin. In fact, it can only exclude places with a signature different from what is found for the object. There is of course also the challenge of determining the local range or signature [37]. Guiterman et al. [38] cited by Watson [39] conclude on their side that Sr isotopic analysis of construction timbers is useful, but on its own can lead to potentially wrong conclusions and that by adding an extra layer of information through ring-width analysis, better archaeological interpretations are possible.

$Sr^+$ often substitutes for $Ca^+$ in chemical compounds. That is how it enters trees as Calcium plays a physiological role in tree growth. Sr is not incorporated in the structure of the tree, but is present as Sr-oxalate crystals in for example the parenchyma cells [32].

**Table 1. Overview of the applications of Sr isotopic analysis on wood in archaeology.**

| Location | Material | Reference |
|---|---|---|
| Chaco | Construction timber | [38–41] |
| Aztalan | Charred vessels | [42] |
| Florida | waterlogged pine | [14] |
| East Mediterranean | archaeological wood (on land and waterlogged) | [43] |
| Heuneburg | waterlogged fir & oak (river) | [44] |

Sr has four different naturally occurring isotopes $^{84}Sr$, $^{86}Sr$, $^{87}Sr$ and $^{88}Sr$. The isotope ratio of interest in geological and archaeological studies is $^{87}Sr/^{86}Sr$. The isotope $^{87}Sr$ forms through the radiogenic decay of $^{87}Rb$ (half-life of 48 billion years) and hence different $^{87}Sr/^{86}Sr$ ratios are found in different geological settings. This difference is determined by the initial Rb/Sr ratio in the deposit and the age of the deposit and hence the accumulation of $^{87}Sr$ over time [45]. Part of the Sr present in bedrocks can be incorporated by living organisms as it is soluble in water, this is called the bio-available Sr. Though the local geology and bedrock are very good predictors of the bio-available Sr it is important to determine the locally available Sr for studies of human mobility and/or provenance of organic material [46]. Bulk rock digestions can lead to very different Sr isotopic compositions from the soil-exchangeable Sr from soils on top of this rock [45]. The soil-exchangeable Sr is defined as the part that can be leached from soil using ammonium nitrate. It mirrors the Sr isotopic composition of the weathered part of the bedrock combined to any other water-soluble Sr sources (such as rainwater, sea spray or dust) [45]. Gneisses form an important part of the bedrock in Sweden, these contain quartz and feldspars that are less susceptible to weathering than other constituents such as biotite and muscovite. Since biotite and muscovite are often relatively enriched in $^{87}Sr$, the bio-available Sr might have a more radiogenic signature than the whole rock [47]. For Denmark, on the other hand, there is no clear link between the Pleistocene sediment cover and the bio-available Sr, however, detailed maps exist facilitating our work [48].

Sampling depth is also especially important in the assessment of bio-available Sr. Where in the soil horizon does a tree gets its Sr from? Poszwa et al. [49] showed that different species have different root depths which leads to different Sr isotopic compositions reflecting the sampling depth. Furthermore, as different species take-up various amounts of Sr, the cycling of Sr is affected by the vegetation [49]. This means that current estimations of bio-available Sr might be biased if the land coverage is very different from the archaeological period under study, or due to other anthropogenic interferences (such as adding lime in agricultural settings). In our case, since we are looking at forested areas that are still forested and with broadly the same species, this should not be a problem. However, it does mean that sampling oaks when studying oaks is to be preferred over sampling other species.

This survey of the literature shows that (i) Sr isotopic analysis can be useful in attempting to provenance timbers that have not been waterlogged, (ii) Sr isotopic analysis of timbers is best interpreted and used in tandem with tree-ring analysis. Therefore, this paper proposes the investigation of three buildings in modern-day Denmark using both dendrochronology and Sr isotopic analysis, in order to assess the suitability of Sr isotopic analysis to provenance timbers that cannot be provenance through dendroprovenancing. Together with these data bio-available Sr is published for a few localities in Western Sweden to help interpret the Sr isotopic compositions of the different timbers.

## Sr isotopic baselines in Scandinavia

In order for Sr isotopic analysis to be able to discriminate between different regions their isotopic signatures need to be different enough. An overview of the Sr isotopic ranges in modern-day Denmark, Sweden and Norway is offered in this paragraph.

**Denmark.** It is especially the work of Frei and Frei [48] that is important for values in Denmark (excluding the Island of Bornholm, which was published separately) they suggest the range $0.7096 \pm 0.0015$ ($2\sigma$) should be used to trace human mobility, and values outside this range indicate non-local origin. This value is determined based on surface waters. For plant and food authenticity they recommend using the slightly lower (but statistically not significantly lower) value of 0.7088 determined through soil leachates and snails [48]. This value will

also be used for the timbers under investigation. In regions of Denmark (like the Island of Fur) where Eocene ash beds are present lower signatures are encountered (0.7038–0.7042), this is especially important when looking at individual timber samples [48]. A tree growing on such a deposit will necessarily have a lower value, outside the 'local' range defined earlier.

**Sweden.** Sweden has a different underlying geology from Denmark, even in the carbonate-rich soils in South-Western Sweden [50]. This of course is especially promising for this particular research question as it ensures a difference in isotopic composition between the two possible regions of provenance (Denmark and Sweden). Within Sweden there are also large differences, allowing for human mobility studies [50]. Blank et al. [50] studied a limited part of Sweden, but one that is of prime importance for the questions tackled here as it covered part of the soils along the Gota river between Gothenburg and Vänern Lake, thought to be the route used to transport timbers from around the lake and river down to the coast where it could be traded.

Sweden lies on the Baltic shield, an old formation reaching from Russia to Norway. The age of the rocks decreases in a south-westerly direction leading to a gradient of Sr isotopic compositions. Eastern Sweden partly lies on the Svecofennian province (1.9±1.75 Ga), western Sweden on the younger Sveconorwegian (sometimes called south-western gneiss) province (1.7 ±0.9 Ga) with the Transscandinavian granite-porphyry belt (TIB, 1.8±1.6 Ga) in between [47, 50, 51]. As a consequence, in broad terms Eastern Sweden has a higher Sr isotopic composition (>0.7300) than (southern and) western Sweden (>0.7220), both of which are much higher than the Danish signatures. But this is of course a very coarse description, igneous intrusions as well as younger sedimentary deposits not eroded during the last glaciation remain in some places (such as Falbygden) [47, 50].

**Norway.** Like South-Western Sweden the southern part of Norway is situated on the Sveconorwegian province of the Baltic Shield [51]. Coastal parts of Norway are affected by sea spray with values similar to the sea signature. The limited Sr isotopic data currently available for Norway reveal a high variability in isotopic composition (0.703–0.715) [52].

## Materials and methods

### Samples from historical buildings

Samples were selected among the dated timbers from three buildings from Jutland in Denmark (Fig 2). The first set of samples is called "Tiendeladen". These oak samples were taken from a house situated on the corner of Tiendeladen 7 and Algade 61 in Aalborg [53]. A single sample from an oak beam is from Brix Gård, also in Aalborg [54]. Pine and oak samples from Nørregade 12 in Horsens [55] were also analysed. These samples reflect what we believed (from the dendrochronological analysis) to be material from Denmark (oak), Sweden (oak) and Norway (pine), as presented below. The aim was to test this hypothesis using Sr isotopes.

### Samples for Sr isotopic baseline along the Gota-river

As mentioned in the introduction we travelled along the Gota-river to sample water, leaves, soil and wood at different locations (Table 2) to determine potential signatures of ancient timbers that would have grown there and would have been floated down to Gothenburg (Fig 2). In Lille Edet we sampled water from the Gota river where it was flowing freely and in a meander where the residence time of the water is higher, this was along a busy road, there were no oaks present so we could not sample any oaks. In Lödöse the water comes from a small stream flowing down the hill and we were able to find oaks to sample. In Torskog we first sampled oaks and soil near the entrance of the forested area, there was no water source in the vicinity. In Torskog 4 there was a river probably feeding one of the nearby lakes which we sampled as

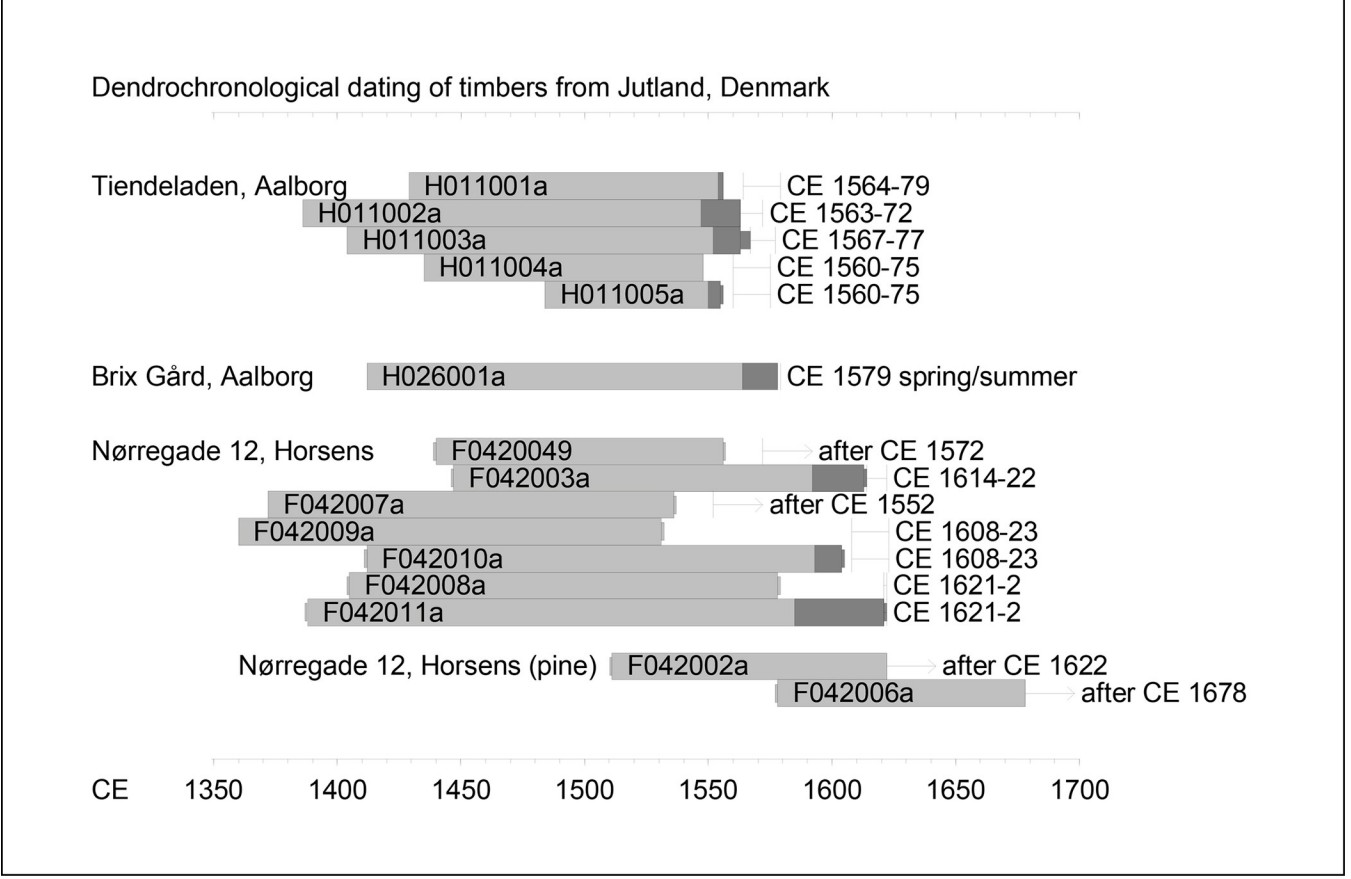

**Fig 2. The dendrochronological dating of the timbers analysed.** The grey bars represent the chronological position of each sample. The dark grey ends represent the sapwood. The line symbols represent the probable date for the felling of each tree, using the appropriate sapwood statistic (for Tiendeladen we used Norway [60] while for Nørregade 12, Horsens we used Germany [18]) (illustration: Aoife Daly).

well as the surrounding soil, no oaks were growing there, so those could not be sampled. Samples were collected on March 10th 2020. Oak tree leaves were brown and collected at the foot of oak trees in forested areas. These are expected to have exchanged Sr with their surroundings during decay.

**Table 2. Sample locations and description in Sweden (see also the map Fig 1).**

| location name | location coordinates | Sample name | Sample type |
|---|---|---|---|
| Lödöse 1 | 58.03647, 12.15606 | LOL | Oak leaves |
| | | LOW | Oak wood |
| | | LOH2O | Water |
| | | LOS | Soil |
| Lille Edet | 58.13223, 12.12212 | LEH2O1 | water in curve of river |
| | | LEH2O2 | water river flowing |
| Torskog 2 | 58.0357, 12.13128 | T2L | young oak leaves |
| | | T2S | Soil |
| Torskog 4 | 58.03268, 12.12509 | T4H2O | $H_2O$ |
| | | T4RS | river soil |
| | | T4S | Soil |

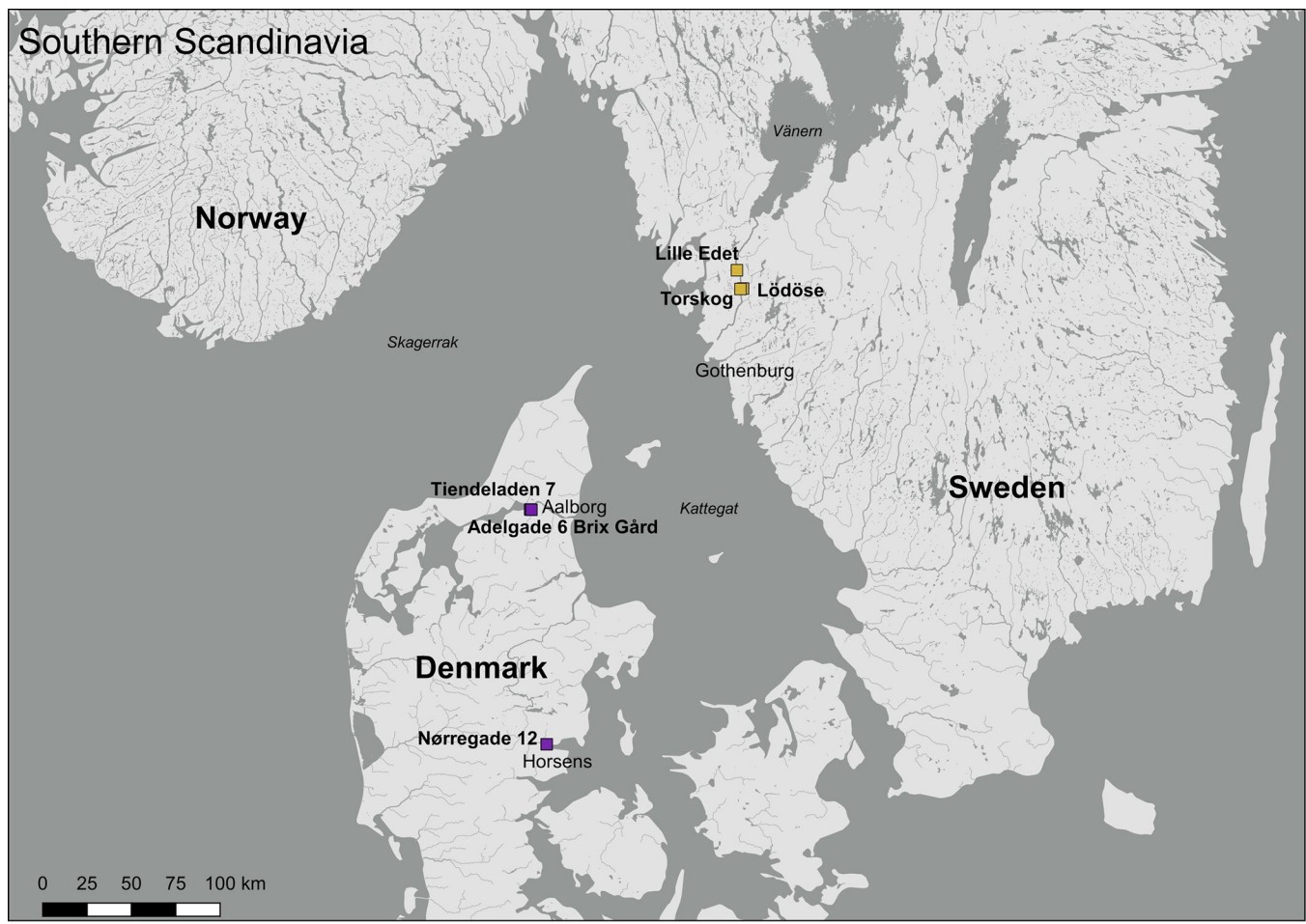

**Fig 1. Map of southern Scandinavia showing the locations of the sites described in the text (baseline samples (yellow squares) and buildings (purple squares)).** The background map is from Natural Earth. Free vector and raster map data @ naturalearthdata.com. Natural Earth (public domain): http://www.naturalearthdata.com. The river data is from www.hydrosheds.org (accessed March 3, 2020). The map is generated using QGIS.org, 2021. QGIS Geographic Information System. QGIS Association.

## Sr isotopic analysis

**Sr leaching and sample digestions.** Water samples (10g precisely measured for concentration determinations) were dried down. Sr was extracted from soils by sonicating 1g of air dried soil in 5ml 1M $NH_3NO_3$ (ammonium nitrate), centrifuging them and removing the supernatant $NH_3NO_3$ using a syringe with a filter mounted on top. This ammonium nitrate is dried down.

For the wood and leaves 500mg of air dried shavings (across different year-rings in the case of timbers) are completely digested by successive treatments in concentrated $HNO_3$ (nitric acid) and a 1:1 mixture of $HNO_3$ and $H_2O_2$ (hydrogen peroxide) at 90˚C. Between each step the material is completely dried down, these steps sometimes need to be repeated a number of times. Care should be taken when adding $H_2O_2$ this produces $CO_2$ (carbon dioxide) and can lead to bubbling over.

**Isolation of Sr through ion chromatography.** Samples were taken up in a 0.5ml of 3M $HNO_3$, spiked with an $^{84}Sr/^{86}Sr = 23.2399$ spike allowing for quantification of the Sr concentration, and then loaded on bio-rad columns filled with 200μL of pre-cleaned mesh 50–100 SrSpec™ (Eichrome Inc./Tristchem) resin. First elements that are not Sr are stripped using

successive 3M $HNO_3$ washes, Sr is eluted by decreasing the pH of the resin and washing with 0.5M $HNO_3$. The collected solution is dried down. If samples are high in Rb (like trees) two consecutive chromatography's can be necessary. When working with organic material the remaining material after elution and drying down is sticky and yellow.

**Thermal ionization mass spectrometry.**   Samples were taken-up in 2.5 μL of a $Ta_2O_5$– $H_3PO_4$–HF (tantalum oxide–phosphoric acid–fluoric acid) activator solution and loaded directly onto previously outgassed 99.98% single Re filaments. Samples were measured at 1250˚C in dynamic multi-collection mode on a VG Sector 54 IT mass spectrometer equipped with eight Faraday detectors (Institute of Geography and Geology, University of Copenhagen). For each sample 10 blocks of 10 measurements were performed, reported errors correspond to within-run (2sd) precisions of the individual runs.

Five ng loads of the NBS 987 Sr standard gave $^{87}Sr/^{86}Sr$ = 0.7102504 ± 0.000010.

**Reagents and blanks.**   For all procedures Ultrapure acids (Seastar™ and dilutions thereof), water from a Milli-Rho-Milli-Q (Millipore) and ultrapure 30% $H_2O_2$ was used. All procedures were performed inside a suite of Class 1000 overpressurized clean rooms, on workstations equipped with Class 100 Hepa filters.

## Dendrochonological analysis

Every tree ring that is preserved on the sample is measured, from the innermost to the outermost, at an accuracy of +/- 0.01 mm. The presence of sapwood and/or bark edge is also recorded. The measurements are made using a measuring stage, in this case one developed by Ian Tyers (Sheffield) using a Heidenhain linear encoder; the analysis utilized Tyers' software DENDRO [56].

For the dendrochronological analysis, the tree-ring widths along single radii of each sample were measured and dating positions were confirmed through statistical correlation (*t*-value Baillie & Pilcher [57]) and replication with extensive tree-ring datasets for Northern Europe, and through visual control of each tree-ring curve. For dating, *t*-values higher than 3.5 are considered significant, but in practice a dating is not accepted unless much higher *t*-values appear with numerous tree-ring datasets at the same dating position (see Tables 3 and 5). When individual tree-ring series from each site match each other with *t*-values higher than 10.0 these are plotted and examined visually, to examine whether timbers might have been made from the same tree. If the tree-ring series display similarity in the longer-term growth trend, alongside the very high *t*-value, they can be evaluated as being probably from the same tree. All measurements are made available in the supporting material (S4 File).

When identifying the provenance of timber through dendrochronological analysis the *t*-value is also utilised. Once the dating of the various samples is found, groupings are identified, by seeing which samples match each other (Table 3). An average for each group is then calculated. The average serves to accentuate the climate signal in the group and diminishes the growth pattern of each individual tree. This average is then tested, at its chronological position, against the large network of tree-ring data for northern Europe. The higher the *t*-value, the greater the agreement between the average and the various datasets. For identifying the provenance of the timber, it is sometimes useful to map the correlations (see e.g. [7, 24, 58, 59]), but here these are given in tables. For identifying the provenance with confidence, it is best to see *t*-values above 9.0, but it is also important to look at the geographical distribution of these correlations, and to be aware of what regions might be lacking relevant tree-ring datasets, when making conclusions (see [6]). (This should not be confused with the *t*-value for identifying whether two samples might come from the same tree. The averaged tree-ring series, both from the site in question, and the site chronologies to which it is being compared, will achieve higher

**Table 3. Correlation (*t*-value Baillie & Pilcher [57]) between the three tree-ring groups from Brix Gård and Tiendeladen and a range of chronologies for oak.** These three groups dendrochronologically indicate a western Swedish provenance.

| Filenames | | Brix Gård H026001a | Tiendeladen group 1 H011M002 snip | Tiendeladen group 2 H011ST4&5 | Site name |
|---|---|---|---|---|---|
| | Dating | AD1412-1578 | AD1404-1563 | AD1435-1555 | |
| Master and site chronologies | | | | | |
| D0137M04 | AD1286-1520 | 9.78 | 8.70 | 4.58 | Denmark, TBT Odense group4, 12 timbers [61] |
| B027oak B | AD1248-1532 | 8.83 | 8.71 | 4.77 | Copenhagen, Gammel Strand B, 21 timbers [27] |
| NB700000 | AD1345-1538 | 8.46 | 10.76 | 5.50 | Denmark, Helsingør (National Museum of Denmark) |
| Ep3mnall | AD1361-1539 | 8.46 | 8.71 | 4.53 | Scotland, Stirling Castle IMPORTS (Crone pers comm) |
| D0134M03 | AD1347-1507 | 7.77 | 6.36 | 3.38 | Denmark, Thomas b Thriges bro, 6 timbers [62] |
| 81272M01 | AD935-1541 | 7.76 | 8.23 | 5.50 | Denmark, Aalborg Boulevarden [64] |
| B027oak C | AD1331-1557 | 7.73 | 10.21 | 7.77 | Copenhagen, Gammel Strand C, 23 timbers [27] |
| 2121M002 | AD1052-1596 | 7.45 | 8.16 | 5.91 | Denmark, Suså Næstved, all posts [64] |
| H009M001 | AD1410-1613 | 7.41 | 8.12 | 6.08 | Denmark, Strandgade Nibe bolværk A6, 3 timbers [65] |
| EP41592 | AD1390-1592 | 7.17 | 7.69 | 6.27 | Scotland, Stirling Castle episode 4 (Crone pers comm) |
| 2M000006 | AD1318-1514 | 7.10 | 7.20 | 5.17 | Denmark, Zealand churches e.g. [66] |
| 8127M001 | AD846-1771 | 6.57 | 7.31 | 4.47 | Denmark, Ålborg, Østerå / Boulevarden, 67 timbers [63, 67] |
| F041M002 | AD1354-1686 | 6.54 | 7.57 | 1.99 | Denmark, Hastrup Mølle, 12 timbers [68] |
| 2x900001 | AD830-1997 | 6.39 | 7.56 | 6.21 | Denmark, Zealand, 227 timbers (National Museum of Denmark) |
| SM100001 | AD1310-1539 | 6.22 | 6.98 | 5.50 | Sweden, Ystad area (Lund University) |
| 4077M001 | AD1310-1540 | 5.99 | 7.11 | 5.53 | Denmark, Nyborg slot [69, 70] |
| B037M001 | AD1337-1550 | 5.99 | 6.97 | 3.35 | Denmark, Favrholm Mølle 20 timbers [71] |
| D014M001 | AD1319-1521 | 5.88 | 5.75 | 3.65 | Denmark, Nørregade Gråbrødrekloster, 2 timbers [72] |
| midtjy17 | AD536-1980 | 5.85 | 6.48 | 3.84 | Denmark, Mid-Jutland (Christensen pers comm) |
| 4077M00X | AD1178-1546 | 5.40 | 6.55 | 5.71 | Denmark, Nyborg Castle, groups A & B, 46 timbers [69, 70] |
| 81M00004 | AD1350-1480 | 5.34 | 5.39 | 1.99 | Denmark, Churches in Vendsyssel, W Sweden group, 24 timbers [73] |
| EP21505 | AD1355-1505 | 5.29 | 5.83 | 2.97 | Scotland, Stirling Castle episode 2 (Crone pers comm) |
| SM000012 | AD1125-1720 | 5.17 | 6.87 | 5.66 | West Sweden [16] |
| B027oak E | AD1315-1663 | 4.67 | 7.09 | 3.43 | Copenhagen, Gammel Strand E oak, 5 timbers [27] |
| H012M001 | AD1379-1576 | 4.54 | 5.50 | 5.07 | Denmark, Aalborg, Sankelmarksgade, 4 timbers [74] |
| SM000005 | AD1274-1974 | 4.54 | 5.27 | 5.47 | Sweden, Skåne / Blekinge (Lund University) |

(*Continued*)

**Table 3.** (Continued)

| Filenames | | Brix Gård H026001a | Tiendeladen group 1 H011M002 snip | Tiendeladen group 2 H011ST4&5 | Site name |
|---|---|---|---|---|---|
| | Dating | AD1412-1578 | AD1404-1563 | AD1435-1555 | |
| 21015M02 | AD1305-1743 | 3.61 | 6.84 | 1.25 | Copenhagen, B&W Site, 24 trees [75, 76] |
| SM100003 | AD1135-1711 | 3.57 | 4.67 | 5.49 | Sweden, Ystad area (Lund University) |
| FTMAS2 | AD1318-1572 | 3.51 | 6.31 | 1.03 | Scotland, Fenton Tower IMPORTS, 5 timbers (Crone pers comm) |
| Chronologies from ships | | | | | |
| Z0923M03 | AD1328-1618 | 9.15 | 9.24 | 6.81 | Sweden, Stockholm, *Vasa* group 3, 31 timbers [11] |
| Z073m001 | AD1385-1574 | 9.10 | 9.46 | 5.80 | Norway, Oslo, Barcode 14 ship, 3 timbers [77] |
| Z141M001 | AD1352-1539 | 8.52 | 10.21 | 6.44 | Klippan 2 shipwreck Gothenburg, 11 timbers [78] |
| q415029m04 | AD1356-1540 | 8.14 | 8.47 | 7.04 | Evangelistas altarpiece, Seville Cathedral, 29 planks [79] |
| Z040M001 | AD1386-1567 | 6.62 | 9.19 | 5.05 | Denmark, Gåsehage Randers, 2 timbers [80] |
| Z173M001 | AD1354-1547 | 6.61 | 6.44 | 4.81 | Norway, Oslo, Bispevika 7 ship, 2 timbers [81] |
| Z249M001 | AD1375-1588 | 6.22 | 7.28 | 5.66 | Norway, Oslo, Bispevika 16, 2 timbers [82] |
| 00652M02 | AD1405-1607 | 6.14 | 6.15 | 4.75 | Copenhagen, B&W Site wreck 2, 2 trees [83] |
| Z157M003 | AD1365-1567 | 5.87 | 7.72 | 6.70 | Norway, Oslo, Bispevika ship 12a, 4 timbers [84] |
| Z119M001 | AD1317-1573 | 5.79 | 6.91 | 4.38 | Norway, Oslo, Barcode ship 4 BC04, 5 timbers [85] |
| Z027M002 | AD1313-1567 | 5.12 | 6.00 | 5.39 | Denmark, Amager Strand ship, 9 timbers [86] |
| Z089m001 | AD1399-1581 | 4.51 | 6.32 | 4.51 | Norway, Oslo, Barcode ship 5, 9 timbers [87] |
| Z043M001 | AD1320-1577 | 4.24 | 5.18 | 6.35 | Germany, FPL 77 4AM wreck, 5 timbers [88] |
| Z109M001 | AD1437-1597 | 2.66 | 5.06 | 5.06 | Norway, Oslo, Barcode ship 1 BC01, 4 timbers [89] |
| Z0923M01 | AD1404-1623 | 2.93 | 4.45 | 5.34 | Sweden, Stockholm, *Vasa* group 1, 63 timbers [11] |
| 006526M1 | AD1357-1578 | 1.28 | 2.06 | 5.04 | Copenhagen, B&W Site wreck 2, 5 timbers [83] |

The grey tone highlights the high *t*-values.

correlations than individual series due to the more robust climate signal that is achieved through greater sample depth in these datasets.)

## Results

### Dendrochronological analysis

The dendrochronological dating of the samples is illustrated in Fig 2.

**Table 4. Matrix of correlation (_t_-value Baillie & Pilcher [57]) between the tree-ring curves of all historical timbers in this study.**

| | Sr sample no. | genus | Dendro sample no. | H011005a | H011004a | H011001a | H011003a | H011002a snip | H026001a | F042003a | F0420049 | F042002a | F042006a | F042007a | F042008a | F042011a | F042009a | F042010a |
|---|---|---|---|---|---|---|---|---|---|---|---|---|---|---|---|---|---|---|
| Tiendeladen | H5 | oak | H011005a | * | 11.0 | 2.71 | 3.30 | 2.18 | 3.20 | 0.06 | - | 1.26 | \ | - | - | - | 1.14 | 0.07 |
| | H4 | oak | H011004a | 11.0 | * | 3.59 | 3.09 | 2.30 | 4.23 | 0.49 | - | 0.62 | \ | - | - | - | 1.00 | 0.02 |
| | H1 | oak | H011001a | 2.71 | 3.59 | * | 3.39 | 3.04 | 5.03 | 1.14 | - | 0.94 | \ | 0.06 | 2.36 | 2.48 | 2.68 | 2.78 |
| | H3 | oak | H011003a | 3.30 | 3.09 | 3.39 | * | 5.68 | 5.33 | 3.38 | 2.16 | - | \ | - | - | - | 0.89 | 0.34 |
| | H2 | oak | H011002a snip | 2.18 | 2.30 | 3.04 | 5.68 | * | 8.00 | 1.96 | 0.70 | 1.03 | \ | 1.67 | 2.21 | 0.81 | 1.46 | 2.10 |
| Brix Gård | X2 | oak | H026001a | 3.20 | 4.23 | 5.03 | 5.33 | 8.00 | * | 2.84 | 1.99 | 0.04 | \ | - | 0.70 | 0.39 | 1.82 | 1.56 |
| Nørregade 12, Horsens | F3 | oak | F042003a | 0.06 | 0.49 | 1.14 | 3.38 | 1.96 | 2.84 | * | 0.98 | 2.15 | - | 0.64 | 0.39 | 0.41 | 1.09 | 1.41 |
| | F4 | oak | F0420049 | - | - | - | 2.16 | 0.70 | 1.99 | 0.98 | * | 0.43 | \ | 0.72 | 0.13 | 0.28 | 0.52 | 0.76 |
| | F2 | pine | F042002a | 1.26 | 0.62 | 0.94 | - | 1.03 | 0.04 | 2.15 | 0.43 | * | 2.40 | \ | - | - | \ | 0.04 |
| | F6 | pine | F042006a | \ | \ | \ | \ | \ | \ | \ | \ | 2.40 | * | \ | \ | \ | \ | \ |
| | F7 | oak | F042007a | - | - | 0.06 | - | 1.67 | - | 0.64 | 0.72 | \ | \ | * | 5.40 | 7.65 | 6.26 | 4.53 |
| | F8 | oak | F042008a | - | - | 2.36 | - | 2.21 | 0.70 | 0.39 | 0.13 | - | \ | 5.40 | * | 16.05 | 7.30 | 7.50 |
| | F11 | oak | F042011a | - | - | 2.48 | - | 0.81 | 0.39 | 0.41 | 0.28 | - | - | 7.65 | 16.05 | * | 9.48 | 8.18 |
| | F9 | oak | F042009a | 1.14 | 1.00 | 2.68 | 0.89 | 1.46 | 1.82 | 1.09 | 0.52 | \ | \ | 6.26 | 7.30 | 9.48 | * | 10.33 |
| | F10 | oak | F042010a | 0.07 | 0.02 | 2.78 | 0.34 | 2.10 | 1.56 | 1.41 | 0.76 | 0.04 | \ | 4.53 | 7.50 | 8.18 | 10.33 | * |

The dash (-) denotes _t_-value less than 0.00. The backslash (\) denotes an overlap less than 30 years. The grey tone highlights the high _t_-values.

**Table 5. Correlation (*t*-value Baillie & Pilcher [57]) between the Horsens timber group in this study that dendrochronologically indicates a Danish provenance.**

| Filenames | | F042M001 | Site name |
|---|---|---|---|
| | Dating | AD1360-1621 | |
| 9M456781 | 109BC-AD1986 | 7.52 | Denmark, Jutland/Funen (National Museum of Denmark) |
| CD60NZ01 | AD1377-1576 | 6.37 | Denmark, Skaføgård, 12 timbers (National Museum of Denmark revised Daly [6]) |
| CD51JZ03 | AD1346-1497 | 5.95 | Denmark, Møllestrømmen, 3 timbers (National Museum of Denmark revised Daly [6]) |
| G008M001 | AD1344-1493 | 5.94 | Denmark, Skodborghus Møllebakken 11 timbers (Christensen pers comm) |
| H137PM01 | AD1408-1555 | 5.89 | Germany, Seeth Haus, 3 timbers (Hamburg Uni revised Daly [6]) |
| CD51MZ01 | AD1364-1585 | 5.78 | Denmark, Gram Bro, 22 timbers (National Museum of Denmark revised Daly [6]) |
| H131YM01 | AD1409-1575 | 5.62 | Germany, Herrenhaus Ostaerrad, 11 timbers (Hamburg Uni revised Daly [6]) |
| H115CM01 | AD1452-1674 | 5.50 | Germany, Preetz Markt 24, 9 timbers (Hamburg Uni revised Daly [6]) |
| CD50PZ01 | AD1285-1482 | 5.45 | Denmark, Varns Klokkehus, 3 timbers (National Museum of Denmark revised Daly [6]) |
| H11ECM01 | AD1368-1502 | 5.33 | Germany, SL St. Johannis Klost, 5 timbers (Hamburg Uni revised Daly [6]) |
| 4077M002 | AD1396-1542 | 5.31 | Denmark, Nyborg Castle, 3 trees [6] |
| H12A1M01 | AD1396-1541 | 5.28 | Germany, Lunden. Hof Eiberg, 5 timbers (Hamburg Uni revised Daly [6]) |
| CD60JZ01 | AD1385-1652 | 5.22 | Denmark, Ulstrup, 3 timbers (National Museum of Denmark revised Daly [6]) |
| G312NZ01 | AD1413-1576 | 5.19 | Germany, Bevern, 3 timbers (Göttingen Uni revised Daly [6]) |
| H11JXM01 | AD1385-1451 | 4.94 | Germany, HL Koberg 2, 6 timbers (Hamburg Uni revised Daly [6]) |
| 4077M003 | AD1418-1546 | 4.90 | Denmark, Nyborg Castle, 2 trees [69, 70] |
| H11HHM01 | AD1379-1531 | 4.87 | Germany, HL Langer Lohberg 47, 14 timbers (Hamburg Uni revised Daly [6]) |
| CD51JZ02 | AD1401-1502 | 4.86 | Denmark, Møllestrømmen, 4 timbers (National Museum of Denmark revised Daly [6]) |
| G330OZ01 | AD1391-1482 | 4.86 | Germany, Hildesheim, 14 timbers (Göttingen Uni revised Daly [6]) |
| CD60OZ01 | AD1370-1588 | 4.85 | Denmark, Bidstrup, 5 timbers (National Museum of Denmark revised Daly [6]) |
| H129JM01 | AD1449-1616 | 4.84 | Germany, Jersbek, 9 timbers (Hamburg Uni revised Daly [6]) |

The grey tone highlights the high t-values.

**Tiendeladen.** The similarity between samples H4 and H5 indicate that they are possibly from the same tree (S1 File). H1, H2, H3 and H5 had sapwood preserved but none of the samples had any bark left (S1 File). Allowing for missing sapwood, using a sapwood statistic for oaks in southern Norway [60], the felling of these trees took place within the period around CE 1567–75. The dendrochronology might suggest two groups of timbers, one including H1, H2 and H3, and another represented by a single tree (H4&H5). Group 1 dates best with material from Gammel Strand C in Copenhagen (*t* = 10.21) and from Helsingør (Elsinore) (*t* = 10.76) and with a wreck found at Klippan, Gothenburg (*t* = 10.21), all of which are believed to be made from material imported from Western Sweden (Table 3). Group 2 also correlates best with Gammel Strand C (*t* = 7.77) and with material from an altarpiece in Seville Cathedral in Spain (Evangelistas altarpiece, Seville Cathedral, *t* = 7.04), which also are

dendrochronologically shown to have come from western Sweden. However, for both these groups the correlation with the few Western Swedish chronologies (i.e. with samples found inside Sweden) is less significant than the correlations with the chronologies from exports (e.g. Sweden, Ystad area, group 1 $t = 6.98$, group 2 $t = 5.50$).

**Brix Gård.** In this case sapwood was preserved until the bark edge, the last ring was incomplete leading to the conclusion that the tree was felled in the spring or summer of 1579 (S2 File). The dendrochronological correlations show highest agreement with material from Odense (TBT group 4, $t = 9.78$), Copenhagen (Gammel Strand B, $t = 8.83$) and Helsingør ($t = 8.46$) (Table 3) but also with the group 3 timber from the Vasa ship ($t = 9.15$) [11] and a ship from Oslo (Barcode 14, $t = 9.10$ [77]) both of which are probably from western Sweden. The Brix Gård timber correlates also well with group 1 from Tiendeladen (for example it achieves $t = 8.00$ with sample H2) (Table 2).

**Nørregade 12, Horsens.** Pine and oak samples from Nørregade 12 in Horsens [55] were dated and provenanced dendrochronologically (S3 File). F9 and F10 are probably from the same tree and F11 and F8 are also probably from the same tree. These four samples, along with F7, form one group dendrochronologically (see Table 4). Using a sapwood statistic for oaks for northern Germany [18] the felling date for these trees is probably around CE 1621–22. Some of these had also previously been investigated through aDNA analysis, which revealed that F10 had haplotype 7 [90]. These five samples best match a chronology for Jutland/Funen ($t = 7.52$) (Table 5).

F3 and F4 form a second dendrochronological group and previous aDNA analysis revealed that F3 had haplotype 1 [91]. The $t$-values for F3 and F4 are smaller than for samples from the other dendrochronological group, F3 best matches two timbers from Gl. Estrup voldgrav ($t = 6.25$) and F4 best matches six timbers from Sostrup ($t = 6.00$) (S3 File). A tentative provenance would be Denmark, but while such comparatively low $t$-values allow the identification of the dating of a timber, a definite provenance identification is difficult.

Both pine samples match with Norwegian sources, F2 matches best with a pine chronology from Oslo Bjørvika B2 k8 [92] ($t = 6.32$) and F6 with Oslo Bispevika B3B7 [43] ($t = 7.67$). The two pine samples do not correlate significantly with each other ($t$-value 2.4) and they probably belong to two separate building phases in Horsens, one phase coinciding with the oak from this building and another towards the last quarter of the 17th century. Even though both pine samples are dendrochronologically matching best with chronologies from the city of Oslo, given that they display low correlation with each other, it is likely that they are from trees of quite different provenances.

As shown in Table 4, no significant correlation appears between the Horsens material and the timbers from the two sites in Aalborg, indicating that these timbers are from a different source.

## Sr isotopes along the Gota-river

**Baseline data.** The results for the baseline values along the Gota-river and of the Gota river itself are given in Table 6. The Sr concentration in leaves is higher than in wood as evidenced by the samples from Lödöse: on the same site a wood sample contained less Sr than the soil. In general, oak tree leaves act as Sr concentrators as both in Torskog 2 and Lödöse leaves contained a higher concentration of Sr (17 times more in the case of Torskog 2 and 4 times more in the case of Lödöse). Whereas rivers or streams contain the lowest concentration of Sr.

The two samples from the Gota river (LEH201 and LEH2O2) have the most radiogenic Sr isotopic signature (a range of 0.7242–0.7246 is obtained when the measurement error is taken into account). This is in line with observations of Frei and Frei [48] that signatures of lake

**Table 6. Results from Sr isotopic determination of samples along the Gota river, Sweden.**

| Location name | Sample name | Sample type | [Sr] (ng/g) | $^{87}Sr/^{86}Sr$ | ±2sd |
|---|---|---|---|---|---|
| Lödöse 1 | LOL | Oak leaves | 9.39 | 0.721204 | 0.000007 |
| | LOW | Oak wood | 1.84 | 0.723626 | 0.000029 |
| | LOH2O | Water | 0.01 | 0.723397 | 0.000035 |
| | LOS | Soil | 2.35 | 0.721769 | 0.000021 |
| Lille Edet | LEH2O1 | Water in curve of river | 0.01 | 0.724546 | 0.000079 |
| | LEH2O2 | Water river flowing | 0.01 | 0.724221 | 0.000034 |
| Torskog 2 | T2L | Young oak leaves | 24.24 | 0.722887 | 0.000128 |
| | T2S | Soil | 1.42 | 0.721166 | 0.000029 |
| Torskog 4 | T4H2O | $H_2O$ | 0.01 | 0.715147 | 0.000029 |
| | T4RS | River soil | 0.07 | 0.723185 | 0.000022 |
| | T4S | Soil | 0.54 | 0.718330 | 0.000008 |

waters in Denmark were slightly enriched in $^{87}Sr$ compared to soil leachates or snails. Though, in their case the difference was not distinguishable statistically. In the case of the Gota-river this can in part be attributed to it's coming from further North in Sweden where run-offs are more radiogenic.

In Lödöse the water sample also has a more radiogenic Sr isotopic signature than both the leaves and soil (which are close to 0.721). The oak wood had a signature closer to the water than to the leaves. This difference might be due to the fact that living wood was taken as opposed to dead leaves lying on the forest floor. The dead leaves probably had leached some of their Sr and some Sr was replaced by less radiogenic Sr contained in rainwater.

In Torskog 4 the signatures of both the water and the soil are significantly lower than all the other values, whereas the soil of the riverbed is more similar to the other sites. We cannot offer an explanation for this.

In broad terms most values range 0.721–0.725, which is significantly different from the values in Denmark.

**Buildings.** Four of the 5 samples from Tiendeladen cluster together $^{87}Sr/^{86}Sr$ = 0.715971–0.716856, only H4 has a slightly higher signature 0.722123 ± 0.000039 (Table 7). The only

**Table 7. Sr isotopic composition of timbers from 3 buildings in Denmark.**

| | [Sr] (ng/g) | $^{87}Sr/^{86}Sr$ | ±2sd |
|---|---|---|---|
| F2 | 3.14 | 0.715594 | 0.000127 |
| F3 | 1.91 | 0.712705 | 0.000008 |
| F6 | 4.15 | 0.712245 | 0.000045 |
| F7 | 25.21 | 0.708565 | 0.000032 |
| F8 | 3.40 | 0.710085 | 0.000076 |
| F9 | 2.20 | 0.709158 | 0.000037 |
| F10 | 2.55 | 0.709365 | 0.000012 |
| F11 | 2.90 | 0.709385 | 0.000037 |
| H1 | 8.17 | 0.716856 | 0.000023 |
| H2 | 2.83 | 0.715971 | 0.000076 |
| H3 | 0.93 | 0.716292 | 0.000049 |
| H4 | 6.56 | 0.722123 | 0.000039 |
| H5 | 2.45 | 0.716472 | 0.00006 |
| X2 | 1.089 | 0.712270 | 0.000004 |

sample from Brix Gård (X2) has a somewhat less radiogenic signature (0.712270 ± 0.000004). For Nørregade 12 Horsens two groups are distinguishable: samples F7-F11 with signatures close to the sea signature (0.708533–0.710161), two samples with more radiogenic values 0.712200–0.712713 and F2 which has the most radiogenic signature and is also associated with the highest uncertainty. Samples F7-F11 were also identified as the same group both dendrochronologically and through aDNA. Samples F3 and F6 though isotopically forming one group cannot be discussed as such since one is an oak sample (F3) and the other one a pine sample (F6).

## Discussion

### Gota River baselines

The difference in isotopic composition between the leaves and the bulk of the wood in Lödöse goes against expectations. Earlier research has shown that this was not the case for cedar wood [93]. This difference can be attributed to a number of factors; (1) analytical problems or errors, it must be noted this is just one experiment, many more samples would be needed to affirm any fractionation between leaves and bulk tree. (2) diagenesis and exchange with the soil: the leaves in Lödöse and Torskog had Sr isotopic signatures almost identical to the soil values. This could be because the trees grow on that soil and hence take-up Sr with this signature (which is the very definition of bio-available Sr) but it could also be that those dead leaves exchanged Sr with the surrounding soil (and less radiogenic rainwater). Wood is extremely susceptible to such exchanges [30] and removing the (sea)water signature has proven impossible [30–32].

Though in Lödöse the river water has a higher signature than the soil, this is not the case in Torskog, so it is impossible to say whether there is a consistent pattern as suggested by Frei and Frei [48] for lakes in Denmark. It is tempting to think that soils would have a lower value because of the mixture with unradiogenic rainwater compared to the rivers draining run-offs containing weathered, more radiogenic minerals, but this seems not entirely valid in our limited experiment.

### Provenance of building timbers

We used the Bayesian likelihood model developed by Hoogewerf et al. [93] to determine the likelihood that the timbers originated from certain regions within Europe. Fig 3 shows the results of these models for 3 different isotopic signatures, representing the range of values in this study. H1 has an uncommon, highly radiogenic signature. Therefore, the Bayesian modelling is able to propose certain regions of provenance with a high probability: either in North-West Spain and Portugal, North-(West) Sweden or southern Finland. F3 has a more common signature, but still spatially limited to particular regions (southern Sweden, mid-west and south west Portugal and Spain, southern Bretagne, parts of central France and Hungary. In the case of F10 which has a signature almost identical to the seawater value, the only conclusion that can be drawn is that is most certainly does not originate from the darker parts of the map (North and Eastern Sweden as well as the North Western parts of Portugal and Spain). It is actually the reverse image from the F3-likelihood map.

### Combining the analysis techniques–dendrochronology and strontium isotopes for provenance determination

**Tiendeladen.** The Sr isotopic analysis provided a number of possible provenances (including parts of Spain/Portugal and Sweden), the dendrochronological results revealed a

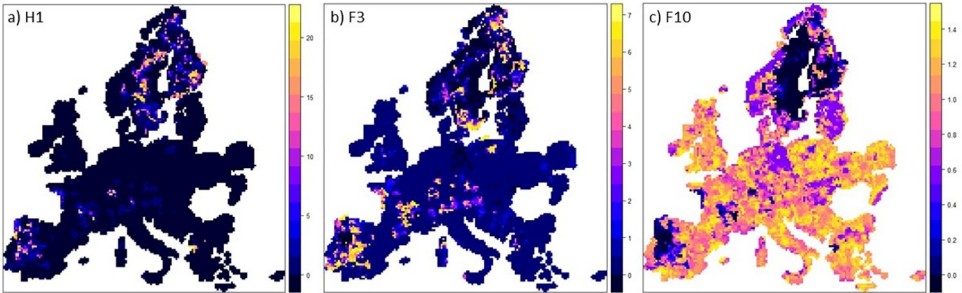

**Fig 3. Statistical probability of samples originating from certain regions in Europe using the R-code developed by Hoogewerff et al.** [93] a) H1 (0.716856 ± 0.000023) from Tiendeladen; b) F3 sample from Horsens (0.712705 ± 0.00008); c) F10 from Nørregade (0.709365 ±0.000012). (R-code used from [93] under a CC BY license, with permission from Elsevier, original copyright 2019).

Southern Swedish provenance. When the information from both techniques is combined it appears that the samples are likely from South-Western Sweden, further south than Gothenburg. This indicates that indeed Denmark was procuring part of its material from Southern Sweden (which at that time was part of the Danish kingdom). Values lower than those upstream of Gothenburg allow one to speculate as to the possibility that wood might have been traded from ports further south such as Varberg. Sample H4, however, has signatures more similar to material from either South-East Sweden or further north upstream from Gothenburg. So either this was procured in a different way than the rest of the wood (i.e. bought from another town) or it was brought to the trading port from further afield. Another possibility that of course cannot be ruled out would be that this tree grew on a local anomaly.

Trade organized from more than one port, including further south, was certainly taking place in the 16th century, and the isotopic analysis provides a strong argument in favour of this hypothesis.

**Brix Gård.** The single sample from Brix Gård has a signature that also matches Southern Sweden, but perhaps further south than the material from Tiendeladen (on account of its less radiogenic signature). Its Sr signature also matches well with sample F3 from Nørregade, but dendrochronologically Brix matches very well with H2 in the Tiendeladen group.

**Nørregade.** In contrast, samples F7-F11 all have signatures close to seawater values thus perfectly overlapping the Danish values. But also matching many other locations across Europe (see Fig 3). Thus, for F7-F11, the provenance based on dendrochronological information is far more useful and the conclusion is that these are materials procured in Denmark. Dendrochronology is actually more precise by even providing the region of origin of samples F7-F11 Jutland or Funen.

For F3, Sr isotopic analysis leads us to conclude it is probably imported from Southern Sweden rather than being local. This explains why it had a much lower *t*-value than the other groups. It fits within the regional chronology for Denmark, but is matching less strongly than the samples from Denmark.

For the two pine samples, first, it must be noted that both values are vastly different, they are within the range of Norwegian values, so the isotopes do not disprove the dendrochronological attribution. In fact, the isotopic analysis provides confirmation that the two pines are of separate provenances, as the dendrochronology also suggested.

Dendrochronology is a powerful tool for identifying the region of origin of historic timber. One of the potential biases in the method is the use of a dataset that in itself has a history of transport and re-use. This phenomenon is clear, for example, when timber particularly from urban centres are from varying sources. When we use these different site chronologies from

the historical dataset, we must look critically at the provenance of the timber in each site chronology, and refrain from grouping the data into larger regional chronologies, until a rigorous examination of how this material correlates is achieved [22]. This necessitates careful interpretation of dendrochronological provenance results in order to avoid circular arguments, and to allow provenance determination to more than a wide, regional level. Using strontium isotopic analysis to interrogate key questions of the dendrochronological dataset an extra level of information is achieved. Based on the results achieved and reported here, a more meticulous assessment of the dataset that is used for identifying provenance dendrochronologically is now possible. The dendrochronology links the dry and wet (unusable for Sr isotopic analysis) datasets through the tree-ring series inter-correlation, and the Sr isotope results guide us to where the diverse dendrochronological groups might be placed.

For Nørregade the groups identified through dendrochronology and aDNA analysis [90] were also found using Sr isotopic analysis.

This analysis now demonstrates that through Sr isotope analysis of dendrochronological samples we can confirm the provenance of traded timber and move towards building 'clean' regional chronologies for more accurate provenance analysis.

## Conclusion

This paper explored the potential added value of combining Sr isotopic analysis with dendrochronological studies. It showed the added value of Sr isotopic analysis on dry building timbers as illustrated in the case study of Brix Gård. Dendrochronology provided more precise provenance for samples F7-F11 than Sr isotopes could. Whereas the analysis of sample F3 prompted a re-examination of the dendrochronological data to a Southern Swedish rather than Danish provenance. Sr isotopic analysis revealed that pines had different Sr-signatures supporting the dendrochronological results of different provenances.

The present analysis also confirmed our hypothesis that wood was traded from ports further south of Gothenburg (founded 1621) along the Western Swedish coast.

A baseline was provided to allow for provenancing of wood along the Gota-river which is thought to be an important water way for the timber trade.

## Supporting information

**S1 File. Dendrochronology report Algade 61 Tiendeladen 7, Aalborg.** Daly, A., 2016. Dendrokronologisk undersøgelse af tømmer fra Algade 61 Tiendeladen 7, Aalborg, NJM 6465. *dendro.dk report* 2016:11, Copenhagen.
(PDF)

**S2 File. Dendrochronology report Brix Gård, Aalborg.** Daly, A., 2020. Dendrochronological analysis of timbers from Brix Gård, Aalborg ÅHM 7235. *Dendro.dk report* 2020:7, Copenhagen.
(PDF)

**S3 File. Dendrochronology report Nørregade 12, Horsens.** Daly, A., 2019. Dendrokronologisk undersøgelse af tømmer fra bygning, Nørregade 12, Horsens (HOM 2393). *dendro.dk report* 2019:5, Copenhagen.
(PDF)

**S4 File. Dendrochronological data.** Tree-ring measurements for all timbers examined in this study.
(FH)

## Acknowledgments

The authors want to thank Toby Leeper for help with the TIMS, Toni Larsen and Cristina Nora for advice on procedures and Tod Waight and Robert Frei for useful discussions.

## Author Contributions

**Conceptualization:** Alicia Van Ham-Meert, Aoife Daly.

**Formal analysis:** Alicia Van Ham-Meert, Aoife Daly.

**Funding acquisition:** Aoife Daly.

**Investigation:** Alicia Van Ham-Meert, Aoife Daly.

**Methodology:** Alicia Van Ham-Meert.

**Supervision:** Aoife Daly.

**Writing – original draft:** Alicia Van Ham-Meert.

**Writing – review & editing:** Aoife Daly.

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
