## [Decision Letter · Decision Letter 0]

12 Apr 2022

PONE-D-22-06131Provenancing 16th and 17th century building timbers in Denmark – combining dendroprovenance and Sr isotopic analysisPLOS ONE

Dear Dr. Van Ham-Meert,

Thank you for submitting your manuscript to PLOS ONE. After careful consideration, we feel that it has merit but does not fully meet PLOS ONE’s publication criteria as it currently stands. Therefore, we invite you to submit a revised version of the manuscript that addresses the points raised during the review process.

Please, read carefully the two reviews and address the comments and suggestions appropriately. Please, make sure that all the steps in your methodology is clearly explained to ensure replication.

We look forward to receiving your revised manuscript.

Kind regards,

Michal Bosela, Ph.D.

Academic Editor

PLOS ONE

Journal Requirements:

"This research was carried out within the project Northern Europe’s timber resource - chronology, origin and exploitation (TIMBER), which received funding from the European Research Council (ERC) under the European Union’s Horizon 2020 research and innovation program (Grant agreement No. 677152). The authors want to thank Toby Leeper for help with the TIMS, Toni Larsen and Cristina Nora for advice on procedures and Tod Waight and Robert Frei for useful discussions."

We note that you have provided funding information. However, funding information should not appear in the Acknowledgments section or other areas of your manuscript. We will only publish funding information present in the Funding Statement section of the online submission form. 

"AD received an starting grant from the European Research Council under the European Union’s Horizon 2020 research and innovation program (Grant agreement No. 677152). Their website is: https://erc.europa.eu/funding/starting-grants.

3. We note that you have referenced (Nordjyske Museer. ÅHM 6465 Algade 61/ Tiendeladen 7, Sted- og lok. nr.: 120516-126, unpublished museum file. Aalborg; 2015.) which has currently not yet been accepted for publication. Please remove this from your References and amend this to state in the body of your manuscript: (Nordjyske Museer. ÅHM 6465 Algade 61/ Tiendeladen 7, Sted- og lok. nr.: 120516-126, unpublished museum file. Aalborg; 2015. [Unpublished]”) as detailed online in our guide for authors

Reviewers' comments:

Reviewer's Responses to Questions

**Comments to the Author**

1. Is the manuscript technically sound, and do the data support the conclusions?

Reviewer #1: Partly

Reviewer #2: Yes

2. Has the statistical analysis been performed appropriately and rigorously? 

Reviewer #1: No

Reviewer #2: Yes

3. Have the authors made all data underlying the findings in their manuscript fully available?

Reviewer #1: Yes

Reviewer #2: Yes

4. Is the manuscript presented in an intelligible fashion and written in standard English?

Reviewer #1: No

Reviewer #2: Yes

5. Review Comments to the Author

Reviewer #1: I have thoroughly read the manuscript submitted by Van Ham-Meert and Daly on identifying the origin of historical timbers from Denmark using dendrochronology and Sr isotopic analysis. The authors present that a combination of both methods provides more detailed information on origin of historical timber constructions. The manuscript presents a timely and scientific important topic. I feel that the topic fits well within the scope of PlosOne, as it is interdisciplinary and quite novel. However, even though the manuscript has great potential, it does not meet the standard for scientific paper, is not well processed and must be distinctly improved before consideration to be published in the journal; therefore, I suggest major revision of the manuscript.

The manuscript does not read well. The chain of arguments is not well developed and some sentences are very complicated and confusing. The identification of sites and samples is very complex. It should be uniform and clear throughout the manuscript. Line numbering or at least page numbering would be very helpful and easier to review the manuscript.

Abstract

- What do you mean with “a number of case studies”. Do you mean the three buildings?

- Why are the last two sentences separated from the other text? Additionally, these two sentences are about what you performed. I would expect that the last sentence should be an outlook or recommendation.

Keywords should be better selected.

- I think that “16th and 17th centuries” are not the best keywords; moreover, you have it in both title and abstract. I would suggest substituting it by “historical construction”, “timber transport” or “cultural heritage”.

The Introduction is too long (almost 8 pages) compared to results (~2 pages) and discussion (~2 pages), includes a lot of redundant information and lacks references. This part should be significantly shortened and thoroughly described. Redundant information should be removed and only relevant text should be preserved so that it is straightforward and follows a line of the story. I would suggest making an outline (what you want to say) and following it.

- There are a lot of different examples which are described in too a detail (e.g. why is it important for this study that the excavations at Gammel Strand where carried out by the Museum of Copenhagen and led by Stuart Whatley… both certainly deserve recognition but this paper is not about their activities…- this redundant information should not be in this paper). I think that only a few examples which are the most relevant for this study should be briefly mentioned with relevant references. If readers are interested, they will find more details in the literature.

- Why do you describe Strontium in detail? I would expect such a detailed description in a paper on biogeochemistry or maybe ecophysiology. Moreover, is it really important to know that it is “just below calcium in the periodic table”? Maybe for chemists or students but definitely not here.

- After the 3rd sentence, you refer to 8 papers but then in the following 2.5 paragraphs you only have two references. However, the references are needed there, for example - there must be various dendro studies which led you to this statement: Through dendrochronological analysis of the remains of timber structures, objects that survive in historic buildings and collections, and that are found through archaeological excavation, a detailed mapping of the sources and destinations for timber has been possible. Another example, I think that you should cite several papers here from various European regions: Master and site chronologies for many regions, from Ireland to Estonia, from Norway to Spain, that have been built from thousands of tree-ring analyses over many decades of dendrochronology in Europe. Or: Extensive tests for Sr isotopic analysis of timbers from shipwrecks and other waterlogged wood were carried out in this project, and by others. Who are the “others”? Again, reference needed. I can find many examples in the Introduction. You should carefully go through it and cite relevant papers or books which you were inspired by.

- “These had never been submerged, hence any Sr in the wood should reflect the geology of the place where these trees originally grew.” How do you know this? You concluded that some timbers were transported. How? In the past, timber was frequently transported on water…

- You say that “Unfortunately, Sr isotopic analysis of waterlogged wood has proven impossible” but then you state “This leads to the inclusion of archaeological timbers that cannot be dated and to innovative techniques being used to complement the picture such as aDNA analysis, elemental and isotopic analysis (O,N,C and Sr).” These are contradictory statements or I did not get your idea…

- “In our case, since we are looking at forested areas that are still forested and with broadly the same species, this should not be a problem.” How do you know that the area was forested in the 16th century?

- I do not understand this sentence “However, it does mean that sampling oaks when studying oaks is to be preferred over sampling other species.” What else do you want sample if you study oaks? I am sorry but this sentence sounds crazy to me.

- I do not agree with this sentence. “Charred vessels offer one very good example of wood material that cannot be provenanced through classical dendroarchaeological means due to the low number of rings.” Charring is not the cause of low number of rings. Charcoal samples can have enough rings for dating.

- “This brief survey of the literature…..” It is NOT brief. It seems to me that you write a review in some paragraphs.

- The main objective and hypothesis of your study should be stated in the last paragraph of the Introduction. You have hypothesis and goals spread out after the Introduction which makes it difficult for the readers to orientate in the text

- I think that the chapter “Sr isotopic baselines in Scandinavia” should be a part of site description in the methods section.

Materials and methods

Methods must be better described! Methodology of dendroprovenancing is missing. I think that numbering of chapters would be very helpful. Identification of samples should be clearly described in the methods. The section is very confusing. You work with recent and historical material but it is NOT clear at all.

- The aim of the study should not be part of the site description.

- Is the site Jylland or Jutland (in figure)?

- How do you know that historical samples are from Denmark, Sweden and Norway? I think that such hypothesis should be tested on living trees where you exactly know the origin of the wood.

- Content of subchapters Tiendeladen, Brix Gard, Horsens are results of dendroprovenancing, aren´t they? It looks like; therefore, it belongs to results, not to methods!

- Table 1 and 3 – What correlation? Pearson? Did you use t-test? Which one? Is the filename important? Why did you do not describe the last column?

- In subchapter Horsens you mention F4 but it is not in Table 2.

- Suddenly, you say what aDNA analysis reveals but you did not describe how you performed the analysis.

- Subchapter Horsens, 2nd paragraph – why do you think that t=6.25 is low?

- Subchapter Horsens, 3rd paragraph – “it is likely that they are from trees of quite different provenances” Why do you think so?

- Subchapter Sr isotopes along the Gota-river – Suddenly you started to describe locations in Sweden but in the previous text you described results! I miss a description of the locations and a map. It is not easy to find orientation without a map.

- I do not understand what you mean: “Water samples were simply dried down.” Did you dry water?

- “these steps sometimes need to be repeated a number of times” Why? How many times?

Results

- Table 5 – It should be better described including identification of samples.

Discussion

- What are the expectations?

- These sentences are redundant “(1) since the leaves were lying on the floor the wood

- and leaves could be from a different tree. This is a poor argument because it would mean that trees separated by only a few meters could have vastly different signatures.”

- “the leaves in Lödöse and Torskog had signatures almost identical to the soil values.” Do you mean Sr values or values of 87Sr/86Sr

- “dead leaves exchanged Sr with the surrounding soil”- I guess you mean rather contamination than exchange.

- Figure 2 is not well described and some letters are too small. What do the percentage and the symbols mean?

- “The cluster of samples from Tiendeladen indicates a provenance in South-Western Sweden, further south than Gothenburg.” Based on what can you say that? I guess that you cannot state so only based on the map in Fig. 2. It is not clear to me how you can exclude other parts of Scandinavia or Portugal and claim that samples are from south-western Sweden. It must be clarified and better explained in the manuscript. Maybe you can say based on the results of both methods…?

- “The single sample from Brix Gård has a signature that also matches Southern Sweden”. How does a signature of the samples match with signature of southern Sweden? Is it just because of similar values? Or any statistics? What is the probability?

- “Dendrochronology is a powerful tool for identifying the region of origin of historic timber. One of the potential biases in the method is the use of a dataset that in itself has a history of transport and re-use” I think that this idea should be better described in the manuscript. Do you think that you can develop a chronology from a mixture of local and imported timber? Do you think it is also possible in the case when you use living oaks from Denmark and historical constructions including oaks from e.g. southern Germany? Do you think that individual tree ring series from distances so far apart would be cross-dated well? Or do you mean biases at smaller geographical areas?

Reviewer #2: Review (anonymous)

Provenancing 16th and 17th century building timbers in Denmark – combining dendroprovenance and Sr

isotopic analysis

Alicia Van Ham-Meert, Aoife Daly

Preliminary remark

Text formatting has unfortunately not been included in this text box. It can be seen in the separately uploaded PDF "Review_Van Ham-Meert_&_Daly_2022.pdf". In this PDF underlining and bold text passages in quotations are from the reviewer.

General statements about the text

The authors focus their research to an important topic in the historical sciences, the

determination of the origin of historical timber (dendroprovenancing). They enter new scientific

territory by extending the conventional procedure in dendrochronology of comparing tree-ring width

curves of the timbers to be determined in their origin with contemporaneous regional tree-ring

calendars by comparing the strontium isotope signatures of the timbers with the Sr signatures of

potential growth areas. This is an innovative approach suitable to test results of

conventional dendroprovenancing with a different methodological approach.

Such multi-proxy analyses have long been used in the natural sciences to explore a scientific

question from different methodological perspectives. A strontium isotope analysis (SIA) in the

context of dendroprovenancing has come to the attention of the reviewer for the first time with

this paper.

In their paper, the authors explain the chemical-technical basics, procedures and limitations

(unusability of waterlogged timbers) of SIA in great detail. This is to be highly commended because

it also introduces readers with less scientific education, for example from the humanities, to the

subject matter, possibilities and limitations of the process. Likewise, the source-critical

comments on the origin of the samples, both of the timber to be examined for its origin and of the

samples from the potential localities of origin (soil, water, wood and leaf samples), which are

repeatedly interspersed in the text, are a benefit for the critical reader. With regard to the

strontium isotope ratios of the soil samples, changes over time, e.g. due to anthropogenic

influences (agriculture), are also addressed. The description of the political conditions and

borders in southern Scandinavia during the period under investigation in the 16th and 17th

centuries CE is also a good contribution to the source-critical historical view of the

investigations presented here. Finally, the discussion of the data in the section "Results -

Baseline data" is exemplary.

Formal suggestions

Because a complementary method to dendroprovenancing is being tested here, this term should also

appear in the keywords.

The years should be supplemented in the title ("Provenancing 16th and 17th century CE ...”) and in

the continuous text by the information "CE" (formerly "AD"), so that the chronological

classification is unquestionable for readers of all disciplines. The religion-related "AD" (anno

domini) should also be replaced in Figure 1 by "CE", which is now common in the natural sciences.

Chemical formulae should be resolved in subsequent brackets when first mentioned in the running

text, e.g. "NH3NO3 (ammonium nitrate)", or listed as a separate list/table at the end of the

article.

It is not clear whether the tree ring width curves of the wood samples are single measurements of

one radius or mean curves from several radii. I think it is methodologically important to specify

this. If one follows the information in the S4-file (S4 file Van Ham-Meert & Daly 2022.fh), which

contains the data of the tree ring widths in the so-called Heidelberg format, it seems to be

exclusively a matter of single radii. If this is the case, I consider this methodologically

questionable as far as the dendrochronological dating of the timbers is concerned. It would also be useful if these tree-ring width data in the *.fh file contained information about the "location". Currently, besides the tree ring widths, they only contain information on dating, number of tree rings, species, pith, sapwood

rings and keycode (see following example).

HEADER:

DateEnd=1556 Length=128 Species=QUSP Pith=1 SapWoodRings=2 KeyCode=H011001A DATA:Tree

149 113 113 99 81 77 67 77 79 53

76 59 69 69 82 49 63 44 70 80

60 73 70 54 103 86 94 75 73 68

83 67 61 56 59 52 53 46 66 72

58 43 45 45 44 63 71 72 56 56

38 57 33 53 67 49 43 49 68 65

49 45 32 29 47 55 52 65 60 54

57 57 78 63 84 92 159 117 146 144

139 143 130 155 168 135 175 151 144 131

89 63 60 120 82 76 133 150 130 113

95 133 112 138 136 88 113 126 115 98

106 115 112 121 114 105 133 123 133 155

104 96 112 80 188 234 262 217 0 0

In the text and in the illustrations, t-values are repeatedly mentioned to express the level of the

dendrochronological correlation of the timbers to each other. Unfortunately, nowhere is it

mentioned whether these values were calculated according to Student (1908), Baillie/Pilcher (1973)

or Hollstein (1980). Also, values are described as significant or non-significant without

explaining which thresholds the authors use for this.

Sometimes the information on the unit of measurement is added to the values without a space ("...

500mg ..."), sometimes there is a space before it ("... 200 μL ..."). The spelling should be

consistent according to the guidelines of PLOS ONE.

Under "Materials and Methods - Sites" it should be mentioned that the sapwood statistics are those

for oaks, and that the Brix Gård site is not mentioned there because it is bark edge wood.

For the Tiendeladen site it should be mentioned that the best correlating chronology for groups 1

and 2 comes from Gammel Strand C, because there are three chronologies from Gammel Strand (B, C and

E) in Table 1. Here, too, significance is mentioned at the end of the section without discussing at

which t-value the authors apply significance ("However, for both these groups the correlation with

the few Western Swedish chronologies (i.e. with samples found inside Sweden) is less

significant.").

For non-dendrochronologists, it can also be confusing when, on the one hand, t-values above 10 are

considered best dates in the text and, on the other hand, samples H4 and H5 are considered to be

from the same stem (stem-matched) because the t-value is above 10.

The heading of Table 1 is "Correlation between the three tree-ring groups in this study that

dendrochronologically indicate a western Swedish provenance." But it is probably not the

correlations between the three tree-ring groups, but the correlations of the individual groups to

the "Master and site chronologies" and the "Chronologies from ships". This should be reworded. It

should also be mentioned in all tables that the values given are t-values according to either

Student (1908) or Baillie/Pilcher (1973) or Hollstein (1980).

In the continuous text, one should consistently repeat the t-values from the tables in brackets and

name the chronologies completely. This makes it easier to understand the argumentation, e.g. at the

Brix Gård site, where it says „The dendrochronological correlations show highest agreement with

material from Odense, Copenhagen [41] and Helsingør (table 1) but also with the group 3 timber from the Vasa ship [8] and a ship from Oslo (Barcode 14 [58]) both of which are probably from western Sweden.”

My suggestion would be: The dendrochronological correlations show highest agreement with material

from Odense (TBT group 4, 9.78), Copenhagen (Gammel Strand B [41], 8.83) and Helsingør (8.46)

(table 1) but also with the group 3 timber from the Vasa ship [8] (9.15) and a ship from Oslo

(Barcode 14 [58], 9.10) both of which are probably from western Sweden.

In Table 3 for the site Nørregade 12, Horsens, a chronology "Jutland/Funen" is mentioned, which

were named "Jylland or Fyn" in the continuous text above ("These five samples best match a

chronology for Jylland or Fyn (table 3)."). For a reader unfamiliar with the geographical

designations of Denmark, such inconsistencies can be irritating. Thus, all the designations in the

Supporting Information should be compared with those in the text.

The heading of Table 4 is "Sample locations and description in Sweden.”. Here I would change the

heading of the fourth column "Sr" to "Sr-sample sources". In the accompanying text, it should be

explained why the sampling of the four individual locations is inconsistent in type and extent

(there are surely good reasons for that).

The chemical formulae used in the section "Sr isotopic analysis - Sr leaching and sample

digestions" I would, as already said in the introduction, either immediately dissolve in brackets,

e.g. "HNO3 (nitric acid)", or dissolve all the formulae used in a table. In this section it also

says "... these steps sometimes need to be repeated a number of times." Why is that?

In the section "Results - "Baseline data", the first paragraph reads "... (17x times more in the

case of Torskog 2 and 4 times more in the case of Lödöse)." The "x" as a multiplication sign is

probably a remnant of an earlier version of the text.

It is a little irritating that the values of the isotope ratios in Table 5 are given with 6 decimal

places, but in the running text only 4 or 3 decimal places are mentioned. „The two samples from the

Gota river (LEH201 and LEH2O2) have the highest Sr isotopic signature (a range of 0.7242-0.7246 is

obtained when the measurement error is taken into account).” If we use Table 5 to calculate the

measurement error (twofold standard deviation), we obtain more precise values (0.724187- 0.724625)

and recognise that the authors have rounded up or down. This is probably due to better clarity.

In Tables 5 and 6, the heading of column 4 should be named "± 2sd" instead of "2sd". This would

make it clearer that a twofold standard deviation is meant here. In Table 5, the country name

"Sweden" should be added to the heading, and in Table 6, "Denmark" should be added: „Table 5:

Results from Sr isotopic determination of samples along the Gota river, Sweden.“, “Table 6: Sr

isotopic composition of timbers from 3 buildings in Denmark.”

In the section "Buildings", a number error has crept into the first line: „Four of the 5 samples

from Tiendeladen cluster together 87Sr/86Sr = 0.175896-0.716879, only H4 has a slightly higher

signature 0.722123 ± 0.000039 (table 6).” It must read “… 87Sr/86Sr= 0.715895-0.716879 …”.

A small mistake has been made in the caption to Figure 2: „Figure 2. Summary of the results of the

strontium isotope analysis for the timber from the three buildings in this study, placed together

with the map reproduced from Hoogewerff et.al. [71], fig 6. (2 column figure)”. Should read “… from

Hoogewerff et al. [71], …)

Figure subheadings or headings (figures, tables) should end consistently either with or without a

full stop.

Proposals regarding content

The claim made in the abstract "By adding the Sr isotopic analysis, a far more detailed

interpretation of the origin of these timbers can be presented" cannot be confirmed in the paper.

The authors themselves contradict this claim in their conclusion when they write, for example:

"Dendrochronology provided more precise provenance for samples F7-F11 than Sr isotopes could.”

The problem that reference curves/master chronologies were often constructed from timbers from

different regions, and that for some historical reference curves it is not possible to know exactly

from which regions the timbers originate, should be made clearer with 2-3 sentences (e.g. Ernst

Hollstein and Bernd Becker have integrated sample material from each other's working area for

certain weakly replicated periods).

The phenomenon of timber transport/trading has not only been a problem of dendrochronology and

dendroprovenancing since the mid-14th century CE. In the chapter "Timber trade in Southern

Scandinavia" it is correctly stated that "Gradually, from around the mid-14th century onwards, wood

and timber in Northern Europe was traded from regions with more abundant forests to regions where

these materials were in high demand and no longer locally available.”. Elsewhere ("Danish buildings

with varied timber sources") it then irritatingly states "From the mid-13th century onwards, oak

was shipped from the regions south and east of the Baltic Sea, in the form of boards of varying

sizes.".

Apart from these incongruent chronological approaches (mid-13th vs. mid-14th century CE), it would

also be desirable for less historically educated readers to point out that extensive timber

transport already took place in the Roman Empire (here Roman Imperial Period to Late Antiquity, c.

27 BCE to mid-6th century CE). There is an extensive literature on this, most recently for example

Bernabei, M., Bontadi, J., Rea, R., Büntgen, U., Tegel, W. (2019): Dendrochronological evidence for

long-distance timber trading in the Roman Empire. PLoS ONE 14(12): e0224077.

https://journals.plos.org/plosone/article?id=10.1371/journal.pone.0224077

One could also briefly mention that supra-regional timber transport in prehistoric times in Europe

up to the Roman period is not a problem for provenance research on timbers: the place where the

timbers processed by humans were found largely corresponds to the place where the trees grew.

This changes temporarily in Roman times, but to my knowledge has not yet been documented for the

subsequent early and high Middle Ages (mid-6th century to mid-13th century CE). From the Late

Middle Ages onwards (mid-13th century CE), the transport of timber in Europe resumed.

The sentence following Table 3, "Using extensive tree-ring datasets across the region, and

examining the highest correlations geographically, the region where the timber achieves highest

correlation can be pinpointed as the region where the tree grew.", I would add to the content that

ecologically similar growth conditions are possible in different regions and can lead to similar

growth curves (tree-ring width data). This is a critical point in dendrochronological provenance

determination (dendroprovenancing), and it should at least be mentioned, because it cannot be ruled

out that comparative data are not available from all regions and that one therefore only recognises

similarities with the areas already researched, which do not necessarily have to be the region of

origin.

The sentence „Hence, we want to use strontium isotopic analysis to be able to pinpoint the source

of the timber, not just where it was used, but where the trees grew.” does not seem logical to me

because the SIA is not supposed to clarify the place of use.

In the section "Sr isotopes along the Gota-river" the colleagues mentioned should also be named:

"This data completes earlier datasets collected by colleagues". Which colleagues are these?

The problem of (undesirable) contamination of oak leaves by rainwater leaching should already be

addressed in the following sentence and not further down in the text: „Oak tree leaves were brown

and collected at the foot of oak trees in forested areas.” In the section "Results - Baseline data"

that follows below, it then states quite correctly “The dead leaves probably had leached some of

their Sr and some Sr was replaced by less radiogenic Sr contained in rainwater.”

In the section “Combining the analysis techniques - …” it says “One of the potential biases in the

method is the use of a dataset that in itself has a history of transport and re-use. This

necessitates careful interpretation of dendrochronological provenance results in order to avoid

circular arguments.” Here it would be appropriate to also refer in the text to the need for the

construction of "clean" regional and also supra-regional chronologies in which the origin of all

wood samples is known. This is, after all, precisely the problem discussed in the essay, which the

SIA is supposed to help with.

In the following sentence „Using strontium isotopic analysis to interrogate key questions of the

dendrochronological dataset an extra level of accuracy is achieved”, I would phrase it more

neutrally like this: “Using strontium isotopic analysis to interrogate key questions of the

dendrochronological dataset an extra level of information is achieved”. The accuracy is not really

that much better, as the authors themselves write in the following "Conclusion". Rather, added

value is achieved through the additional safeguarding with an independent proxy, which, however, is

only possible for dry woods. (see passages underlined below): „It showed the added value of Sr

isotopic analysis on dry building timbers as illustrated in the case study of Brix Gård.

Dendrochronology provided more precise provenance for samples F7-F11 than Sr isotopes could.

Whereas the analysis of sample F3 prompted a re-examination of the dendrochronological data to a

Southern Swedish rather than Danish provenance. For the pine samples the dendrochronology

identified them as originating from two locations in Norway. Sr isotopic analysis confirmed that

the two pines had different provenances”. I would make the last part of the sentence, in bold, more

neutral: „Sr isotopic analysis revealed that

pines had different Sr-signatures supporting the dendrochronological results of different

provenances.”

6. PLOS authors have the option to publish the peer review history of their article (what does this mean?). If published, this will include your full peer review and any attached files.

Reviewer #1: No

Reviewer #2: No

---

## [Author Response · Author response to Decision Letter 0]

13 Oct 2022

Dear reviewers,

Thank you very much for your work. Please find attached a full answer to each of your points.

Kind regards,

The authors

---

## [Decision Letter · Decision Letter 1]

9 Nov 2022

PONE-D-22-06131R1Provenancing 16th and 17th century CE building timbers in Denmark – combining dendroprovenance and Sr isotopic analysisPLOS ONE

Dear Dr. Van Ham-Meert,

Thank you for submitting your manuscript to PLOS ONE. After careful consideration, we feel that it has merit but does not fully meet PLOS ONE’s publication criteria as it currently stands. Therefore, we invite you to submit a revised version of the manuscript that addresses the points raised during the review process.

 Please, see and address a few minor suggestions provided by the reviewer.

We look forward to receiving your revised manuscript.

Kind regards,

Michal Bosela, Ph.D.

Academic Editor

PLOS ONE

Journal Requirements:

Reviewers' comments:

Reviewer's Responses to Questions

**Comments to the Author**

1. If the authors have adequately addressed your comments raised in a previous round of review and you feel that this manuscript is now acceptable for publication, you may indicate that here to bypass the “Comments to the Author” section, enter your conflict of interest statement in the “Confidential to Editor” section, and submit your "Accept" recommendation.

Reviewer #2: (No Response)

2. Is the manuscript technically sound, and do the data support the conclusions?

Reviewer #2: Yes

3. Has the statistical analysis been performed appropriately and rigorously? 

Reviewer #2: Yes

4. Have the authors made all data underlying the findings in their manuscript fully available?

Reviewer #2: Yes

5. Is the manuscript presented in an intelligible fashion and written in standard English?

Reviewer #2: Yes

6. Review Comments to the Author

Reviewer #2: A big thank you to the authors for following my suggestions so understandingly and unpretentiously!

General remarks:

I still disagree with you that measuring a single radius of tree-ring widths is sufficient. It may be sufficient for a secure dating in the cases presented here, and I trust completely that the authors have secured their dating very well. But for basic statistical considerations, at least a second control measurement should always be made for tree-ring width measurements. This may sound "oldschool", but it is important and remains correct. I do not expect any changes in your text in this respect, but I would be pleased if you would take my advice to heart and implement it in the future as a contrast to a "one radius quick & dirty" dendrochronology.

In my review I wrote:

"It is a little irritating that the values of the isotope ratios in Table 5 are given with 6 decimal places, but in the running text only 4 or 3 decimal places are mentioned. <the from="" gota="" river="" samples="" the="" two=""> If we use Table 5 to calculate the measurement error (twofold standard deviation), we obtain more precise values (0.724187-0.724625) and recognise that the authors have rounded up or down. This is probably due to better clarity."

And you answered:

"Yes, this is what we did, we are unsure whether you would like this to be changed?"

And I say to you: No, no change necessary. The shortening of the decimal places serves the clarity of the text! Leave it as it is.

In your conclusion you wrote:

"For the pine samples the dendrochronology identified them as originating from two locations in Norway. Sr isotopic analysis confirmed that the two pines had different provenances."

And I suggested that the second sentence be worded more neutrally:

"Sr isotopic analysis revealed that pines had different Sr-signatures supporting the dendrochronological results of different provenances."

It is a pity that you could not come to terms with this. But leave it as it is. It is your statement.

However, I have two things to note that I insist on changing:

1. In the file "S4 file Van Ham-Meert & Daly 2022.fh" the location information is still missing, although the reply to the reviewer says: "Done: location info inserted in tree-ring dataset file." Have I perhaps received an old version of the file here?

2. In table 7, column 4, the "±" sign was forgotten: "±2sd" instead of "2sd". Please correct.

Best regards

Your reviewer</the>

7. PLOS authors have the option to publish the peer review history of their article (what does this mean?). If published, this will include your full peer review and any attached files.

Reviewer #2: No

---

## [Author Response · Author response to Decision Letter 1]

15 Nov 2022

Dear reviewer,

We are grateful for your kind and encouraging words as well as for your impartiality. The manuscript is greatly improved by your attentive reading.

Kind regards

The authors

---

## [Editor Report · Decision Letter 2]

18 Nov 2022

Provenancing 16th and 17th century CE building timbers in Denmark – combining dendroprovenance and Sr isotopic analysis

PONE-D-22-06131R2

Dear Dr. Van Ham-Meert,

We’re pleased to inform you that your manuscript has been judged scientifically suitable for publication and will be formally accepted for publication once it meets all outstanding technical requirements.

Kind regards,

Michal Bosela, Ph.D.

Academic Editor

PLOS ONE
---

## [Editor Report · Acceptance letter]

6 Dec 2022

PONE-D-22-06131R2 

Provenancing 16th and 17th century CE building timbers in Denmark – combining dendroprovenance and Sr isotopic analysis 

Dear Dr. Van Ham-Meert:

I'm pleased to inform you that your manuscript has been deemed suitable for publication in PLOS ONE. Congratulations! Your manuscript is now with our production department. 

Kind regards, 

on behalf of

Dr. Michal Bosela 

Academic Editor

PLOS ONE